# Neuropilin-1 and Integrins as Receptors for Chromogranin A-Derived Peptides

**DOI:** 10.3390/pharmaceutics14122555

**Published:** 2022-11-22

**Authors:** Angelo Corti, Giulia Anderluzzi, Flavio Curnis

**Affiliations:** 1Faculty of Medicine, Università Vita-Salute San Raffaele, 20132 Milan, Italy; 2Tumor Biology and Vascular Targeting Unit, Division of Experimental Oncology, IRCCS San Raffaele Scientific Institute, 20132 Milan, Italy

**Keywords:** chromogranin A, vasostatin-1, catestatin, angiogenesis, tumor diagnosis, neuropilin-1 integrin avβ6, integrin avβ8

## Abstract

Human chromogranin A (CgA), a 439 residue-long member of the “*granin*” secretory protein family, is the precursor of several peptides and polypeptides involved in the regulation of the innate immunity, cardiovascular system, metabolism, angiogenesis, tissue repair, and tumor growth. Despite the many biological activities observed in experimental and preclinical models for CgA and its most investigated fragments (vasostatin-I and catestatin), limited information is available on the receptor mechanisms underlying these effects. The interaction of vasostatin-1 with membrane phospholipids and the binding of catestatin to nicotinic and b2-adrenergic receptors have been proposed as important mechanisms for some of their effects on the cardiovascular and sympathoadrenal systems. Recent studies have shown that neuropilin-1 and certain integrins may also work as high-affinity receptors for CgA, vasostatin-1 and other fragments. In this case, we review the results of these studies and discuss the structural requirements for the interactions of CgA-related peptides with neuropilin-1 and integrins, their biological effects, their mechanisms, and the potential exploitation of compounds that target these ligand-receptor systems for cancer diagnosis and therapy. The results obtained so far suggest that integrins (particularly the integrin avb6) and neuropilin-1 are important receptors that mediate relevant pathophysiological functions of CgA and CgA fragments in angiogenesis, wound healing, and tumor growth, and that these interactions may represent important targets for cancer imaging and therapy.

## 1. Introduction

Human chromogranin A (CgA), a member of the “*granin*” protein family, is a 439-residues long protein present in the secretory vesicles of various normal and neoplastic neuro-endocrine tissues and neurons, and exocytotically released into the blood stream upon cell stimulation [1,2].

Abnormal levels of CgA, detected by immunoassay, are present in the blood of patients with neuroendocrine tumors or with other diseases, such as cardiovascular, gastrointestinal, renal, and inflammatory diseases [3].

CgA undergoes various post-translational modifications in different cells and tissues, including phosphorylation, sulphation, glycosylation, and proteolytic cleavage [2,4]. Intra-granular and/or extra-cellular proteolytic enzymes, such as furin, cathepsin L, prohormone convertase 1 and 2, thrombin and plasmin, can cleave the full-length CgA precursor (CgA_1-439_) at different sites to generate various biologically active fragments involved in the regulation of the innate immunity [5,6,7,8], cardiovascular system [9,10,11,12], metabolism [13,14,15], angiogenesis [16,17,18], tissue repair [19] and tumor growth [14,20,21]. These fragments include N-terminal large polypeptide fragments (e.g., CgA_1-373_) [17], as well as shorter fragments, such as CgA_1-76_ (vasostatin-1) [9], CgA_79-113_ (vasoconstrictive-inhibitory factor) [22], CgA_1-113_ (vasostatin-2) [23], CgA_250-301_ (pancreastatin) [13], CgA_352-372_ (catestatin) [24], CgA_411-436_ (serpinin) [25,26], and others [3,14]. The in vitro and in vivo assays used to investigate the biological effects of all these fragments and their mechanisms are reviewed in detail elsewhere [2,6,10,12,14,18,26].

Marie Hélène Metz-Boutigue and coll. were the first to demonstrate that vasostatin-1 and catestatin, two of the most investigated fragments, are endowed of antibacterial and antifungal activities [2,6,27,28]. However, several studies have shown that these peptides can also affect the physiology of mammalian cells and exert several regulatory functions under physiological and pathological conditions. For example, catestatin and vasostatin-1 induce vasodilation [2,9,29]. In addition, catestatin inhibits nicotinic-cholinergic-stimulated catecholamine secretion [24], promotes the release of histamine from rat mast cells and stimulates monocyte chemotaxis [30]. Furthermore, vasostatin-1, catestatin, and full-length CgA_1-439_ reduce myocardial contractility and relaxation [31,32,33], counteract the β-adrenergic-stimulated positive inotropism, and regulate the coronary tone [12]. Additionally, CgA and vasostatin-1 can affect, in an opposite manner, the adhesion of cardiomyocytes, keratinocytes, fibroblasts, and smooth muscle cells to proteins of the extracellular matrix [2,34]. CgA and vasostatin-1 can also prevent the disassembly of vascular endothelial cadherin-dependent adherens junctions [35], inhibit vascular leakage induced by tumor necrosis factor-α [35], and exert angiogenic effects [17], whereas catestatin and CgA_1-373_ promote angiogenesis [16,17]. In human microvascular endothelial cells, vasostatin-1 inhibits the expression of tumor necrosis factor-α–induced intercellular adhesion molecule-1, the release of monocyte chemoattractant protein-1, and the relocation of high mobility group box-1 [36]. Physiological concentrations of full-length CgA_1-439_, and vasostatin-1 may also have a regulatory role in wound healing [19] and tumor growth [21,37,38], and exert several other biological effects in the regulation of metabolism and cardiovascular system [14].

Despite the numerous activities reported for CgA, vasostatin-1, and catestatin, limited information is available on the underlying receptors. Biochemical studies have shown that vasostatin-1 can interact with phosphatidylserine and other membrane-relevant phospholipids [39]. Furthermore, a mechanism involving the binding of vasostatin-1 to heparan sulfate proteoglycans and phosphoinositide 3-kinase-dependent eNOS phosphorylation has been observed in bovine aortic endothelial cells [40]. Other studies have shown that the nicotinic acetylcholine receptor mediates the inhibitory effect of catestatin on the secretion of catecholamines from chromaffin cells [14,24]. Catestatin can also act on the β2-adrenergic receptor, as suggested by the results of a combination of experimental and computational studies [41]. Interestingly, recent studies have shown that integrins and neuropilin-1 may also act as important receptors for CgA, vasostatin-1, CgA_1-373_, and other fragments, in endothelial cell biology, cardiovascular function, angiogenesis, wound healing, and tumor growth. Here, we review the structural requirements for the interactions of CgA and CgA-fragments with neuropilin-1 and integrins, their biological effects, their mechanisms, and the potential use of compounds targeting these ligand-receptor interactions for cancer diagnosis and therapy.

## 2. Neuropilin-1 as a Receptor for Chromogranin A-Derived Peptides

### 2.1. Biological Effects of CgA and Its Fragments in Angiogenesis and Tumor Growth

Studies in various pre-clinical models of solid tumors have shown that systemic administration of recombinant CgA_1-439_ to tumor-bearing mice enhances the endothelial barrier function, inhibits tumor neo-vascularization, and reduces tumor growth [17,21,35]. In vitro studies have shown that CgA_1-439_ is an anti-angiogenic molecule and that an anti-angiogenic site is located in the region 410–439 (i.e., the C-terminal region of the full-length protein). A latent (or less active) site is also present in the N-terminal region 1–76, this site requiring vasostatin-1 liberation by proteolytic cleavage of the Q_76_K_77_ peptide bond for full activation [17]. Physiological concentrations of CgA_1-439_ and vasostatin-1 inhibit, with U-shaped dose response curves, the pro-angiogenic activity of fibroblast growth factor-2 and vascular endothelial growth factor, two important proteins involved in angiogenesis regulation [17,42]. Studies on the mechanism of action have shown that the anti-angiogenic and anti-tumor activity of CgA_1-439_ depends on the induction of protease-nexin-1 in endothelial cells, a serine protease inhibitor endowed with potent anti-angiogenic activity [21]. Furthermore, a recent study, aimed at elucidating the pro-angiogenic mechanisms triggered by the fragment CgA_1-373_, have shown that cleavage of the R_373_–R_374_ dibasic site of circulating CgA in tumors and the subsequent engagement of neuropilin-1 by the fragment are crucial mechanisms for the spatio-temporal regulation of angiogenesis in cancer lesions and, consequently, for the regulation of tumor growth [37]. Opposite to CgA_1-439_ and vasostatin-1 (anti-angiogenic), the fragment CgA_1-373_ can promote angiogenesis with a bell-shaped dose-response curve and with a maximal activity at 0.2–1 nM, i.e., at concentrations found in certain cancer patients [17]; thus, the full-length CgA_1-439_ and its fragments may form a balance of anti- and pro-angiogenic factors that can be finely regulated by proteolytic cleavage at Q_76_ and R_373_. Studies in murine models of lung carcinoma, mammary adenocarcinoma, melanoma and fibrosarcoma have shown that circulating CgA can be partially cleaved in tumors after the R_373_ residue, and that the consequent exposure of the PGPQLR_373_ site is crucial for tumor progression [37]. A blockade of the exposed PGPQLR_373_ site with specific polyclonal and monoclonal antibodies (unable to recognize the CgA precursor) reduced tumor vascular bed, blood flow, and tumor growth [37]. These findings suggest that cleavage of CgA by proteases and the subsequent exposure of the PGPQLR_373_ site may contribute to regulate the vascular physiology in tumor tissues [37]. Given that no CgA was produced by cancer cells in the models studied, it is very likely that these fragments were generated by cleavage of bloodborne CgA in tumor lesions. It appears, therefore, that CgA molecules present in the blood can work as an “*off*/*on*” switch for the activation of angiogenesis in tumors after local cleavage.

### 2.2. Mechanisms Underlying the Biological Effects of CgA Fragments in Angiogenesis and Tumor Growth

The findings described above raise a series of questions. First, which proteases can switch on this pro-angiogenic mechanism in tumors? Second, which receptor mediates the biological activity of CgA_1-373_? Considering that thrombin and plasmin are known to be activated in tumors, and that these enzymes can efficiently cleave the R_373_R_374_ peptide bond [17,43], both enzymes are good candidates for cleaving CgA in tumors. Although other proteases might also be involved (discussed below), it is interesting to note that full-length CgA_1-439_ can induce, in endothelial cells, the production of protease-nexin 1 (a potent inhibitor of plasmin, plasminogen activators, and thrombin [21]), and that the plasminogen activator inhibitor-1 inhibits CgA cleavage to CgA_1-373_ by cultured endothelial cells [43]. It is therefore tempting to speculate that changes in the relative levels of these protease/anti-protease molecules in tumor lesions represent a major mechanism for the regulation of the CgA-dependent angiogenic switch.

Regarding the second question, the results of biochemical studies suggest that CgA_1-373_ can bind to neuropilin-1 (NRP-1) with high affinity (Kd = 3.49 ± 0.73 nM), via its C-terminal PGPQLR_373_ sequence [37]. The functional role of this ligand-receptor interaction is suggested by the fact that the pro-angiogenic effects of this fragment are blocked by anti-neuropilin-1 or anti-PGPQLR_373_ antibodies. No interaction of full-length CgA and CgA_1-372_ with neuropilin-1 occurs, indicating that the PGPQLR_373_ binding site is cryptic in the full-length precursor and that the C-terminal arginine residue (R_373_, which is absent in CgA_372_) is necessary for the binding [37].

Studies on the topology of the NRP-1-binding site showed that CgA_1-373_ and short peptides containing the PGPQLR sequence interact with a pocket of the *b1* domain of the receptor, a site that recognizes peptides and polypeptides ending with the so-called C-end Rule (CendR) motif (R/K-X-X-R/K, as in the prototypical CendR peptide RPARPAR). Remarkably, this binding pocked can also accommodate and bind the C-terminal sequence of VEGF_165_, a pro-angiogenic factor that contains a CendR motif (CDKPRR) [44,45]; thus, CgA_1-373_, VEGF_165_, and even the short PGPQLR and RPARPAR peptides, all with a C-terminal arginine, compete for the same binding pocket of the *b1* domain of NRP-1. Although the PGPQLR_373_ sequence cannot be fully considered a CendR motif, as it lacks the first R/K residue of the consensus sequence, it is interesting to note this sequence, such as the CendR motif, ends with an arginine that is crucial for NRP-1 recognition.

The importance of the C-terminal arginine of CgA_1-373_ for neuropilin-1 recognition is also supported by the results of molecular docking and molecular dynamics experiments, performed with CgA_352-372_ (catestatin) and CgA_352-373_ (catestatin-R). Despite these compounds differ only for the presence of an arginine (C-terminal sequence: PGPQL in catestatin; PGPQLR in catestatin-R), a clear difference in the interaction with NRP-1 was observed [46]. The interaction of catestatin-R with neuropilin-1 showed strong similarity with that of the compound EG00229, a small inhibitor of NRP-1 that contains an arginine with a free carboxyl group and whose structure in the complex with NRP-1 has been resolved by crystallography studies. In both cases, complex formation is driven by salt bridges between the guanidine moiety of the C-terminal arginine of the ligand and the carboxyl group of an aspartate residue of neuropilin-1 (D_48_) [46]. Remarkably, despite the presence of other positively charged arginine residues and amino-groups in catestatin-R, the best binding mode was obtained with the interaction of the C-terminal R_373_ of the ligand with D_48_ of the receptor.

Another important question raised by these findings concerns the possible involvement of co-receptors. Experimental evidence showed that the pro-angiogenic activity of CgA_1-373_ in assays based on endothelial cell spheroids can be inhibited by mecamylamine and α-bungarotoxin, two antagonists of nicotinic acetylcholine receptors [37]. Considering that (a) the nicotinic acetylcholine receptors are expressed on endothelial cells and are known to contribute to the regulation of angiogenesis [47,48,49,50], and (b) catestatin (CgA_352-372_) is known to bind nicotinic acetylcholine receptors [51,52,53], this class of receptors may represent important co-receptors for CgA_1-373_ signaling in endothelial cells. It cannot be excluded, however, that other receptor systems are also involved.

A final point that should be discussed concerns the issue of counterregulatory mechanisms. Experimental evidence suggests that R_373_ is rapidly removed from CgA_1-373_ by plasma carboxypeptidases, when this fragment is released in circulation [37]. Given the importance of R_373_ for neuropilin-1 recognition, the cleavage of CgA_1-373_ to form CgA_1-372_ may represent an important mechanism to limit the CgA_1-373_ activity at the site of its production (for example, in cancer lesions) and to avoid systemic effects.

Thus, the results obtained so far support a model in which cleavage of the R_373_R_374_ bond of circulating CgA, followed by neuropilin-1 engagement in tumors, and the subsequent removal of R_373_ in plasma, represent a sort of “*off*/*on*/*off*” switch for the spatio-temporal regulation of angiogenesis in tumor lesions (see Figure 1A for a schematic representation of the model).

Interestingly, it is well known that pro-hormone convertases can cleave proteins at dibasic sites (R/K-R/K), and that carboxypeptidase H/E remove the C-terminal R or K. It is possible that also these enzymes are brought into play in the regulation of the CgA-dependent angiogenic switch. Indeed, it is possible that a lower expression, or a reduced activity, of carboxypeptidase H/E in tissues in which CgA_1-373_ is overproduced (see below) may contribute to activate the pro-angiogenic switch. On the other hand, the normal expression/function of carboxypeptidase H/E in other tissues might have a role in the generation of the anti-angiogenic vasostatin-1 fragment (by removal of K_77_ after cleavage of the K_77_/K_78_ dibasic site) and CgA_1-372_ (by removal of R_373_ after cleavage of the R_373_/R_374_ dibasic site), thereby promoting an anti-angiogenic effect, a hypothesis that deserves to be tested.

### 2.3. Potential Therapeutic Applications of Compounds That Interfere with CgA Fragment/Neuropilin-1 Interaction

The unbalanced production of anti- and pro-angiogenic factors in tumor tissues can trigger aberrant angiogenesis and altered vascular morphology, which, in turn, may contribute to tumor cell proliferation, invasion, trafficking and formation of metastases [54,55,56]. The studies performed in patients with multiple myeloma have shown that CgA is cleaved into the proangiogenic form CgA_1-373_ in the bone marrow and that, consequently, the ratio of pro-/anti-angiogenic forms of CgA is higher in patients compared to healthy individuals [43]. Enhanced CgA cleavage correlated with increased levels of vascular endothelial growth factor and fibroblast growth factor-2 in the bone marrow plasma, and with an increased bone marrow microvascular density [43]. Studies on the mechanism of action revealed that multiple myeloma and endothelial cells can promote CgA cleavage through the activation of the plasminogen activator/plasmin system [43].

Other studies aimed at evaluating the extent and prognostic value of CgA cleavage in patients with pancreatic ductal adenocarcinoma, an aggressive cancer arising from the exocrine component of the pancreas [57,58,59], have shown that cleavage of the R_373_R_374_ bond and of other sites in the C-terminal region of circulating CgA is increased in these patients. Remarkably, CgA cleavage predicts progression-free survival and overall survival in these cancer patients [38]. Experimental evidence, obtained using various pre-clinical models of pancreatic ductal adenocarcinoma, suggests that the plasminogen activator/plasmin system has a role in CgA processing in this case, and that CgA cleavage has a functional role of in the regulation of tumor vascular biology. Remarkably, anti-PGPQLR_373_ antibodies capable of blocking the binding of CgA_1-373_ to neuropilin-1 can reduce the growth of pancreatic ductal adenocarcinoma in mice, which implicates an important role of neuropilin-1 as mediator of these effects [38].

As cleavage of plasma CgA in tumors and the consequent interaction with neuropilin-1 may represent an important mechanism for the regulation of tumor vascular biology and growth, the assessment of the extent of CgA fragmentation in cancer patients may have a prognostic value, whereas the development of compounds that target and block this ligand-receptor interaction (e.g., anti-PGPQLR_373_ monoclonal antibodies) may have a therapeutic value (see Figure 1B for a schematic representation of this concept).

### 2.4. Role of CgA Fragment/Neuropilin-1 Interactions in Cardiovascular Regulation

Global neuropilin-1 null mice develop severe cardiovascular abnormalities, indicating that neuropilin-1 has also a crucial cardiovascular function [60]. In addition, the observation that the selective knockout of neuropilin-1 in cardiomyocytes and vascular smooth muscle cells leads to cardiomyopathy, increased propensity to heart failure, and reduced survival after myocardial infarction, suggests a role for neuropilin-1 in the pathogenesis of cardiovascular diseases [61]. Based on these notions, and on the fact that CgA is the precursor of various cardio-regulatory fragments, a recent study has investigated the possibility that the fragment CgA_1-373_ affects the myocardial performance by interacting with neuropilin-1 [46]. Hemodynamic assessment (performed using the Langendorff rat heart model) and studies on the mechanism of action (performed using perfused hearts and cultured cardiomyocytes) have shown that CgA_1-373_ can elicit negative inotropism and vasodilation, whereas no significant effects were observed with CgA_1-372_, which lacks the C-terminal arginine necessary for neuropilin-1 recognition [46]. These effects were abolished by antibodies directed against the PGPQLR_373_ sequence of CgA_1-373_. Furthermore, ex vivo and in vitro studies showed that these biological effects are mediated by the endothelium and involve neuropilin-1, Akt/NO/Erk1,2 activation and S-nitrosylation [46]. The effects elicited by CgA_1-373_ and the lack of activity observed with CgA_1-372_ suggest that CgA_1-373_ is a cardio-regulatory factor and that the removal of its C-terminal arginine by carboxypeptidases may work as an important switch for “*turning off*” its cardio-regulatory activity.

## 3. Integrins as Receptors for CgA and CgA-Derived Peptides

### 3.1. The Interaction of CgA and CgA Fragments with Integrins

The first evidence for a role of integrins as receptors for full-length CgA and vasostatin-1 comes from a study on wound healing in injured mice [19]. This study has shown that CgA and vasostatin-1, at nanomolar concentrations, selectively interact with the integrin αvβ6 (see Table 1), suggesting that both polypeptides are natural ligands of this integrin. The integrin αvβ6 is an epithelial-specific cell-surface receptor of vitronectin, tenascin, fibronectin, and also of the latency associated protein of TGFβ1 [62,63,64]. In general, αvβ6 recognizes a site consisting of an arginine-glycine-aspartate (RGD) motif, followed by the LXXL/I motif (RGDLXXL/I) [65,66]. The latter motif folds into one α-helical turn upon binding to the receptor [65,66,67,68,69,70]. Interestingly, a short CgA-derived peptide comprising the residues 39–63 (CgA_39-63_, FETLRGDERILSILRHQNLLKELQD) is sufficient for high-affinity binding and highly selective recognition of αvβ6 (Ki: 15.5 ± 3.2 nM, Table 1) [71]. This peptide exhibits a degenerate RGDLXXL/I motif, in which a glutamate residue (E_46_) is present in place of the leucine downstream of the RGD sequence (RGDEXXL). Interestingly, in this peptide both the RGD motif (CgA_43-45_) and the adjacent sequence (CgA_46-63_) are crucial for αvβ6-integrin binding affinity and selectivity, as suggested by the observation that the replacement of RGD with RGE abrogates integrin recognition (Table 1), and the deletion of even a part of the C-terminal sequence markedly reduces binding affinity and selectivity [19]. The molecular determinants of αvβ6 recognition by CgA_39-63_ have been elucidated by NMR, computational, and biochemical studies [71]. Homonuclear and heteronuclear multidimensional NMR analyses of this peptide in physiological conditions have shown that the region between residues E_46_ and K_59_ has an α-helical conformation, while the RGD motif is relatively flexible; the first three turns of the α-helix are amphipathic, with the hydrophilic aminoacid residues E_46_, R_47_, S_50_ on one side and the hydrophobic I_48_, L_49_, I_51_, L_52_ on the opposite side [71]. The propensity of CgA_39-63_ to form an α-helix is consistent with the results of a previous NMR study on CgA_47-66_, an antifungal CgA-derived peptide, showing all-helical conformation in trifluoroethanol, an α-helix-promoting solvent [72]. Saturation transfer difference (STD) spectroscopy experiments, performed with the extracellular region of human αvβ6 and isotopically labeled (^13^C/^15^N) CgA_39-63,_ have shown that the hydrophobic residues I_48_, L_49_, I_51_, and L_52_ of the α-helix display the strongest STD values (>75%) [71], suggesting that these aminoacids contribute to receptor binding. Molecular docking experiments led to a model of receptor-ligand interactions that is highly reminiscent of that proposed for the proTGFβ1/αvβ6 complex [71].

No binding of CgA_39-63_ has been observed to other integrins (such as α1β1, α6β4, α3β1, α9β1 α6β7, α5β1, αvβ3, αvβ5, and αvβ8) at low-nanomolar concentrations [19]. However, peptides of containing the CgA_39-63_ region could recognize the integrin αvβ3 and other integrins of the RGD-family when used at high concentrations in the low-micromolar range [19]. For example, competitive binding assays performed with purified integrins showed that peptide CgA_39-63_ can bind αvβ6 and αvβ3 with Ki values of 15.5 and 2192 nM, respectively (Table 1), indicating that this peptide can recognize both integrins, but with markedly different affinities [71].

### 3.2. Role of CgA/Integrin Interactions in Wound Healing

The αvβ6 integrin is barely expressed in normal adult tissues, whereas it is highly expressed during wound healing, tissue remodeling, and embryogenesis [73,74]. This integrin is involved in TGFβ1 maturation, it regulates the expression of matrix metalloproteases and modulates keratinocyte adhesion, proliferation, and migration in wound healing [19,62,64,75]. It is possible, therefore, that CgA and its fragments have also a role in the regulation of the wound healing process, by interacting with this integrin. According to this view, experimental data showed that local injection of recombinant CgA_1-439_, but not of a CgA_1-439_ mutant with RGE in place of RGD, can accelerate wound healing in mice [19]. Immunohistochemical analysis of skin tissue sections obtained from injured mice, showed that CgA, but not the RGE mutant, could induce keratinocyte proliferation and thickening of epidermis, suggesting that CgA can regulate the keratinocytes physiology and the process of wound healing through an RGD-dependent mechanism that likely involves the αvβ6-integrin. Interestingly, both CgA and αvβ6 are expressed in wound keratinocytes [19,76]. The fact that both ligand and receptor are expressed at injured sites lends further support to the hypothesis that the CgA/αvβ6 interaction may have a pathophysiological role in this process.

Regarding the integrin αvβ3, this cell-adhesion receptor is an important player in endothelial cell biology and angiogenesis [77,78]. Although it is unlikely that this integrin has a receptor function for the circulating CgA polypeptides, considering its micromolar affinity, significant ligand-receptor interactions can possibly occur at sites where CgA is produced and, therefore, where this protein is present at high concentrations, such as in the microenvironment of wound keratinocytes and neuroendocrine secretory cells or in the microenvironment of neuroendocrine tumors. Furthermore, this interaction might occur on αvβ3-positive endothelial cell after the interaction with other high-affinity binding sites, i.e., through a sort of ligand-passing mechanism.

### 3.3. Potential Diagnostic and Therapeutic Applications of CgA-Derived Peptides That Interact with Integrins in Cancer

The integrin αvβ6 is overexpressed by several types of cancer cells, such as head and neck squamous cell carcinoma, pancreatic ductal adenocarcinoma, breast, colon, liver, and ovarian cancers, and others [73,74,79,80,81,82,83]. This integrin modulates cancer cell invasion, inhibits apoptosis, and, importantly, is involved in the maturation of TGFβ1, a potent immunosuppressive cytokine. Increased expression levels of αvβ6 are prognostic indicators of poor survival in patients with various types of tumors [79,82,84,85,86], and various ligands of this integrin coupled to tumor imaging agents are currently being tested in cancer patients for tumor imaging purposes [87,88,89,90,91]; thus, the development of CgA-derived peptides capable of recognizing this integrin in tumors is of great experimental and clinical interest. Following this line of thought, experimental work has been carried out to obtain new peptides with higher affinity for αvβ6-integrin, starting from CgA_39-63_ as a lead compound. The model of CgA_39-63_/αvβ6 interactions, obtained by NMR and computational studies, allowed to predict that restoring the canonical RGDLXXL motif by replacing the glutamate (E) residue in the RGDERIL site of CgA_39-63_ with a leucine (L) may increase its affinity for αvβ6. Intriguingly, the replacement of E_46_ with L not only increased, as expected, the binding affinity for αvβ6, but, unexpectedly, also that for the integrin αvβ8 (Table 1); thus, the E_46_L replacement converted CgA_39-63_ into a bi-selective ligand of both αvβ6 and αvβ8 integrins (Ki: 1.6 ± 0.3 nM and 8.5 ± 3.7 nM, respectively) integrins [71]. Chemical “stapling” of the α-helix of the E_46_L-CgA_39-63_ mutant, by side-chain-to-side-chain cross linking with a triazole-bridge, further increased the affinity for both αvβ6 and αvβ8 (Ki: 0.6 ± 0.1 nM and 3.2 ± 1.2 nM, respectively) by stabilizing the α-helix [71]. Notably, the αvβ8 integrin represents another cell-surface receptor expressed by various carcinoma cells [92,93,94]; thus, the mutated/chemically stapled peptide (called peptide **5a**) represents a strong bi-selective ligand for these integrins, which can be potentially exploited as a tumor-homing ligand for delivering imaging and anticancer compounds to αvβ6/αvβ8 single- or double-positive tumors, such as oral and skin squamous cell carcinoma [95] (see Figure 2 for a schematic representation of this concept). This hypothesis is supported by the results of a very recent study aimed at investigating the tumor-homing properties of compounds consisting of peptide **5a** coupled with IRDye 800 CW (a near-infrared fluorescent dye) or with ^18^F-NOTA (a label for positron emission tomography) [96]. This study showed that both conjugates can bind αvβ6 and αvβ8 with an affinity similar to that of the free peptide and that they can selectively recognize various αvβ6/αvβ8 single- or double-positive cancer cells, including cells from melanoma, pancreatic carcinoma, oral mucosa, prostate, and bladder cancer. Furthermore, biodistribution studies, performed with these conjugates in mice bearing orthotopic or subcutaneous αvβ6-positive pancreatic tumors, showed high target-specific uptake of fluorescence- and radio-labeled peptide by tumors [96]. Tumor-specific uptake of the fluorescent conjugate was also observed in mice bearing αvβ8-positive prostate tumors [96], confirming the hypothesis that peptide **5a** can home to αvβ6- and/or αvβ8-positive tumors.

Remarkably, both αvβ6 and αvβ8 integrins (which are upregulated in many tumors and, in the case of αvβ8, also in tumor infiltrating Treg cells) can activate the latency associated peptide/TGFβ complex, through interactions of integrins with the RGD sites of the complex [73,83,92]. These interactions can lead to the local activation of TGFβ in the tumor microenvironment, a potent immunosuppressive mechanism that may contribute to tumor progression. Interestingly, in vitro studies have shown that peptide **5a** can inhibit the integrin-mediated TGFβ activation [96]. Thus, in principle, the peptide **5a** can be used not only as a ligand for delivering imaging or anticancer agents to αvβ6/αvβ8 single- or double-positive tumors, but also as a tumor-homing inhibitor of these TGFβ activators.

Finally, considering the role of αvβ6/αvβ8-mediated TGFβ activation in fibrosis [97] the dual targeting capability of peptide **5a** might be also exploited in the development of anti-fibrotic drugs. This is another hypothesis that deserves to be investigated.

## 4. Conclusions

The results obtained so far suggest that integrins (particularly the integrin αvβ6) and neuropilin-1 are important receptors that mediate relevant pathophysiological functions of CgA and its fragments in angiogenesis, wound healing, and tumor growth. Experimental evidence indicates that these interactions may also represent important targets for cancer imaging and therapy. Although further work is necessary to clarify the receptor mechanisms of CgA and its fragments in the regulation of cardiovascular homeostasis, metabolism, and tumor growth, the results obtained so far highlight the complexity of the “CgA system”, which consists of a multitude of CgA-derived peptides and various receptors. The complexity of this system is even higher if we consider that full-length CgA and some of its fragments show biphasic dose-response curves in angiogenesis assays, as well as in cardio-regulatory and tumor pre-clinical models, likely because of the activation of counterregulatory mechanisms at higher doses. These mechanisms are not clearly understood and, therefore, their full elucidation remains a challenge. A third level of complexity is related to the fact that CgA undergoes differential post-translational modifications in different cells and tissues, such as glycosylation, sulfation, and phosphorylation. As most of the studies carried out so far on the biological functions of CgA have been performed with recombinant or synthetic peptides lacking these modifications, the impact of these structural modifications on proteolytic cleavage, fragment generation, receptor recognition, and biological activity, remains to be investigated.

## Figures and Tables

**Figure 1 pharmaceutics-14-02555-f001:**
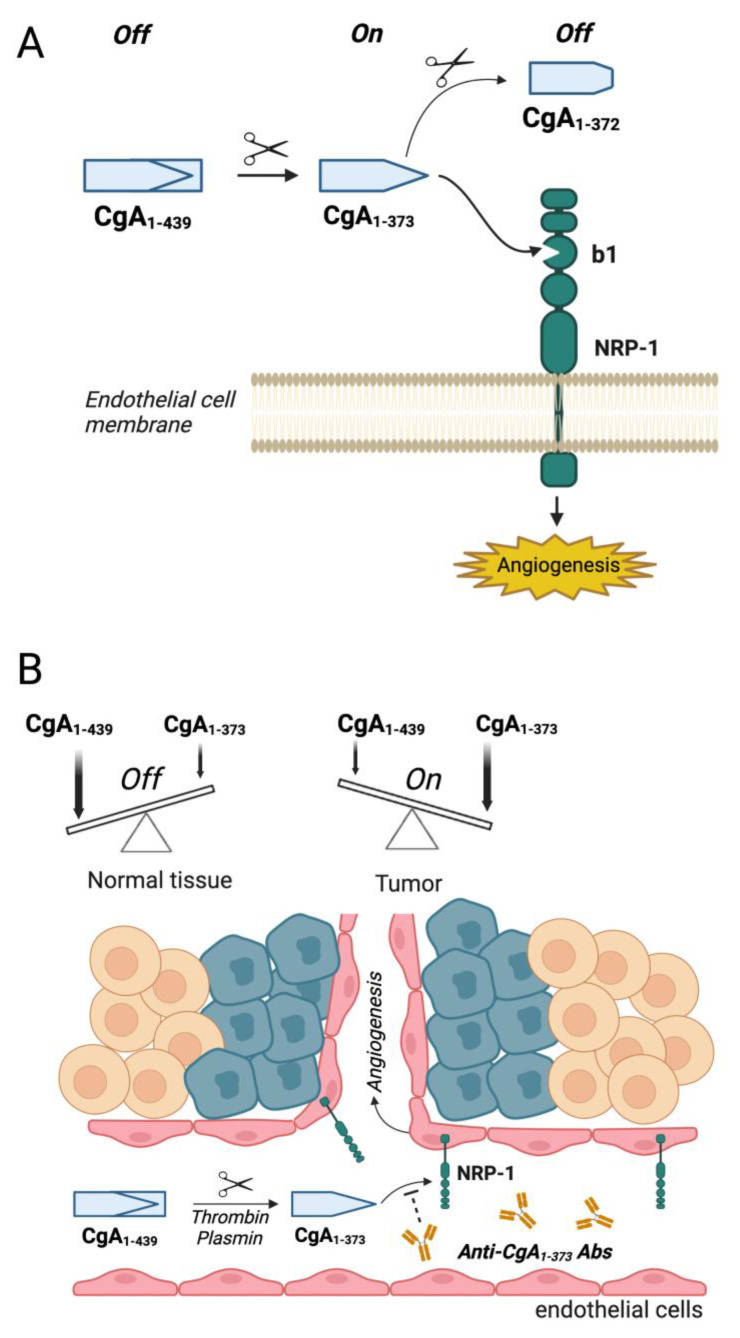
Hypothetical model of the CgA-dependent “*off*/*on*/*off*” switch for the regulation of angiogenesis in tumor and its inhibition by anti-PGPQLR antibodies. (**A**) Mechanisms of activation/deactivation of the NRP-1 binding site of chromogranin A. According to this model cleavage of the R_373_R_374_ peptide bond of full-length CgA (CgA_1-439_) leads to exposure of the PGPQLR sequence, a site that can recognize the CendR-binding pocket of the *b1* domain of neuropilin-1 (NRP-1) on endotal cells. Removal of the R_373_ residue by carboxypeptidases causes loss of NRP-1 recognition [37]. (**B**) Mechanism of anti-tumor activity of anti-PGPQLR antibodies. Cleavage of bloodborne full-length CgA (CgA_1-439_) in tumors, e.g., by plasmin or thrombin, causes loss of anti-angiogenic CgA_1-439_ and generates the pro-angiogenic CgA_1-373_ fragment, which may interact with NRP-1 and contribute to promote angiogenesis and tumor growth. Antibodies against the NRP-1 binding site of CgA_1-373_ (anti-PGPQLR antibodies) block the CgA_1-373_/NRP-1 interaction and, consequently inhibit angiogenesis and tumor growth [37]. This schematic representation has been prepared using the BioRender software.

**Figure 2 pharmaceutics-14-02555-f002:**
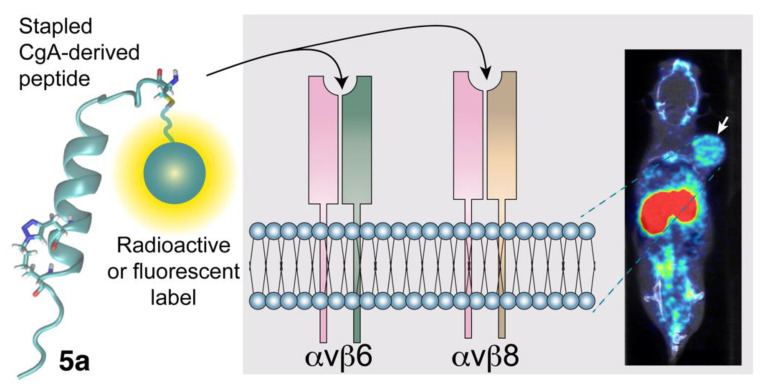
Use of the CgA-derived peptide **5a** (stapled) for delivering imaging or therapeutic compounds to αvβ6/αvβ8 single- or double-positive tumors. The peptide **5a**, derived from the region 38–63 of human CgA (originally published in [96]) is characterized by the sequence CFETLRGDLRILSILRX_1_QNLX_2_KELQ, where X_1_ and X_2_ are propargylglycine and azidolysine residues, respectively, which form a triazole bridge after a click chemistry reaction, thereby increasing the α-helix stability. Peptide **5a** can be exploited for delivering radioactive or fluorescent imaging compounds to αvβ6/αvβ8 single- or double-positive tumors or for developing new therapeutic tumor-homing agents. The image in the right panel shows the radiotracer uptake in a mouse bearing a pancreatic tumor implanted subcutaneously (arrow), as assessed by PET/CT scan (originally published in [96]).

**Table 1 pharmaceutics-14-02555-t001:** Binding affinity of CgA-derived fragments for integrins.

Competitor	Competitive Binding Assay to Integrins (Ki, nM) ^a^					Ref.
	αvβ6	αvβ8	αvβ3	αvβ5	α5β1	
CgA_1-439_	105 ± 34	>2000	>2000	>2000	>2000	[19]
Vasostatin-1	74 ± 30	>10,000	>10,000	>10,000	>10,000	[19]
CgA_39-63_	15.5 ± 3.2	7663 ± 1704	2192 ± 690	3600 ± 525	9206 ± 1810	[71]
CgA_39-63_ (RGE)	>50,000	>50,000	>50,000	>50,000	>50,000	[71]
CgA_39-63_ (RGDL)	1.6 ± 0.3	8.5 ± 3.7	1928 ± 226	2405 ± 592	924 ± 198	[71]
CgA_39-63_ (RGDL)-Stapled	0.6 ± 0.1	3.2 ± 1.2	2453 ± 426	2741 ± 615	1310 ± 389	[71]

^a^ *Ki*, equilibrium dissociation constant of the competitor (mean ± SEM). The *Ki* values were determined by competitive binding assay using an isoDGR-peroxidase conjugate as a probe for the integrin binding site [71].

## Data Availability

Not applicable.

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
