# Peer review of "Neuropilin-1 and Integrins as Receptors for Chromogranin A-Derived Peptides"

_pharmaceutics, 2022, doi:10.3390/pharmaceutics14122555_

Round 1

Reviewer 1 Report

Manuscript Number: pharmaceutics-2000542

General comments:

This is an excellent review article written by an expert in the field of Chromogranin A, cancer, and angiogenesis. The authors very nicely put forth neuropilin-1 and integrins as receptors for chromogranin A-derived peptides. This will be a well-cited review article. The readers will benefit tremendously if the authors address the following points:

Major points:

1.    Since the article discusses a lot of clinical findings, it would be best if the authors change the “experimental models” to “pre-clinical models” throughout the MS.

2.    Page 2, line 1: Instead of adding citations at the end of the sentence, it would be best for the readers if the authors cite the original articles at the end of each functional items as follows … the regulation of innate immunity (citation), cardiovascular system (citation), metabolism (citation), angiogenesis (citation), tissue repair (citation) and tumor growth (citation).

3.    Page 2, line 2&3: Instead of writing “lacking the C-terminal region (e.g., CgA1-373)”, please change to “N-terminal large polypeptide fragment (e.g., CgA1-373)”.

4.    Page 2, line 3-5: Please cite the original papers after each peptide/function: CgA1-76 (vasostatin-1) (citation), CgA79-113 (vasoconstrictive-inhibitory factor) (citation), CgA1-113 (vasostatin-2) (citation), CgA250-301 (pancreastatin) (citation), CgA352-372 (catestatin) (citation), CgA411-436 (serpinin) (citation)…

5.    Please cite the original papers after each function associated with a peptide in this paragraph.

6.    The major focus of this review article is angiogenesis and cancer. Below are several key points:

a.    The full-length human Chromogranin A (CgA1-439) increases endothelial barrier and decreases neovascularization.

b.    In vitro, CgA1-439 inhibits angiogenesis and that anti-angiogenic domain resides in CgA410-439 (i.e., R-serpinin).

c.     While vasostatin 1 (CgA1-76) inhibits angiogenesis, large N-terminal polypeptide (CgA1-173) promotes angiogenesis. Furthermore, R-serpinin (CgA410-439) inhibits angiogenesis. Therefore, the pro-angiogenic switch hinges upon the removal/keeping of 3 amino acids: (i) removal of Q76 (Q75¯Q76), (ii) keeping R373 (R373¯R374), and (iii) keeping R410 (R409¯R410). Please mention which of the enzymes are involved for the above cleavages. The well-established enzymes that are involved in proteolytic cleavage are as follows: pro-hormone convertases cleave at the dibasic site (R/K¯R/K); carboxypeptidase H/E (CPH/CPE) removes the N-terminal R or K; peptidyl glycine a-amidating monooxygenase amidates C-terminal glycine after the removal of R or K by carboxypeptidase H/E. So, CPH/CPE expression must be low, or function must be diminished to generate CgA1-373. This may be true in several carcinomas where CgA1-373 is overexpressed. The anti-angiogenic switch on the other hand relies on the removal/keeping of 3 amino acids: (i) keeping Q76 (Q76¯K77), (ii) removal of R373 (L372¯R373), and (iii) removal of R410 (R410¯R411). The normal expression/function of CPH/CPE will favor anti-angiogenic switch. This is further complicated by the protease nexin-1 and plasminogen activator inhibitor 1 (PAI-1). Please rewrite this part considering the above comments.

7.    The authors can coin the term catestatin-R to describe CgA352-373.

Reviewer 2 Report

In this review manuscript, the authors present an elaborate summary of the biological functions of CgA-related peptides, their interactions, and mechanisms thereof with neuropilin-1 and integrins receptors. More importantly, they discuss the potential exploitation of compounds that target these ligand-receptor systems for cancer diagnosis and therapy. This is an interesting and important research field, and this comprehensive review should be helpful for researchers to understand the importance and the latest progression of CgA-related peptides for therapeutic development.

In general, although well-written and comprehensive, this review is packed with a lot of information and thus goes short on clear readability. I have suggested breaking some sections down into subsections and adding figures and tables.

I recommend the acceptance of this manuscript after the following points are considered.

1. For section 2.1, a table introducing all the possible effects of CgA fragments, how the effects are studied (i.e. assays used), and the quantification of the effect would be nice to provide.  

2. Section 2.1 seems too lengthy and packed with a lot of information. It is advisable to further break it down into 2 different sections –

Biological effects of CgA fragments/neuropilin-1 interactions in angiogenesis and tumor growth

Mechanisms of biological activities of CgA fragments

The same applies to the integrin section.

3. It would be very helpful to include a second table where all the important mechanisms through which various CgA fragments operate for clear readability.

4. Full forms of CgA have been used in some cases even after the introduction of the short form. Please correct.

5. In the table mentioned above in (3), it would be nice to add techniques and how they are used (in short) to assess the mechanism.

6. How is Figure 2 relevant to provide in this review? I could see many other figures such as a visual summary of the role of CgA in various therapies, and potential diagnostic and therapeutic applications, which will be way more important, and relevant and are required to provide in this review.
